# Evaluation of Geriatric Sarcopenia and Nutrition in the Case of Cachexia before Exitus: An Observational Study for Health Professionals

**DOI:** 10.3390/geriatrics7050102

**Published:** 2022-09-21

**Authors:** Titus David Moga, Ioana Moga, Monica Sabău, Alina Cristiana Venter, Dana Romanescu, Erika Bimbo-Szuhai, Lavinia Mihaela Costas, Anca Huniadi, Diana Maria Rahota

**Affiliations:** 1Departmen of Morphological Disciplines, Faculty of Medicine and Pharmacy, University of Oradea, 410028 Oradea, Romania; 2Department of Psycho Neuroscience and Recovery, Faculty of Medicine and Pharmacy, University of Oradea, 410028 Oradea, Romania; 3Department of Medical Disciplines, Faculty of Medicine and Pharmacy, University of Oradea, 410028 Oradea, Romania; 4Department of Surgical Disciplines, Faculty of Medicine and Pharmacy, University of Oradea, 410028 Oradea, Romania

**Keywords:** BIA assessment, clinical nutrition, sarcopenia, cachexia

## Abstract

It is important to assess the physical and nutritional status of the body using a bioelectrical impedance analyzer (BIA) in patients with cachexia; however, the correlation between cachexia and nutritional evaluations remains unclear. The objective of this study is to follow the effects of diet therapy in patients with cachexia/sarcopenia, using parameters measured by BIA, clinical parameters, and other nutrition-related assessments in patients with osteoporosis. This study aims to clarify the correlation between BIA-measured parameters, clinical parameters, and other nutrition-related assessments. Methods: Measurements of body composition, a clinical assessment of the sarcopenia/cachexia, and nutritional goal setting/a nutrition care process were performed. Results: The number of subjects was 200, of which 15 people (7.5%) were diagnosed with sarcopenia/cachexia. Univariate analyses showed that participants with a high body-fat mass tend to develop sarcopenic obesity (*p* = 0.029), those who lost a significant and progressive amount of muscle mass tend to develop sarcopenia (*p* = 0.001), as well as those with malnutrition (*p* < 0.001). The regression study shows not only the correlation but also the cause of the correlation, as is the case with obesity. As obesity increases, so does the sarcopenic index (this can explain sarcopenic obesity), and as fat mass decreases it leads to muscle mass loss, increasing the risk of cachexia with age. Conclusions: There was an improvement, but statistically insignificant, in cachexia and the nutritional objectives (*p* > 0.05); at the same time, correlations were established between the independent parameters (sex, age) and malnutrition parameters (hemoglobin and amylase) with the parameters of the research.

## 1. Introduction

A rapidly aging society warrant continued advances in conventional medical care [1,2]. As frail, older adults should have access to comprehensive healthcare and nursing care, providing comprehensive care using multicomponent assessments is imperative [2]. There are very few national studies [3,4] that have examined the degree of sarcopenia in elderly patients. The process of muscle mass loss begins after the age of 40 but is more accelerated after the age of 60–65 [5]. One of the most pronounced changes in the elderly is the loss of mobility and physical capacity, thus deteriorating their quality of life. These changes occur because of the progressive loss of skeletal muscle mass and function, a process known as sarcopenia [6,7]. Studies focus on cachexia in the context of malignant diseases [8,9], and in 2022, Baker Rogers described it as an associated disease next to a basic one, correlated with anorexia, insulin resistance, and an accentuated pro-inflammatory process [10]. In the current study, cachexia is associated with osteosarcopenia, and the status and risks of developing cachexia are evaluated. Sarcopenia, a key driver of frailty [11], is a syndrome characterized by the presence of both muscle mass and reduced muscle function due to aging, inactivity, malnutrition, and conditions such as cachexia [12]. Recently, some studies have described sarcopenic dysphagia (dysphagia due to sarcopenia of the whole body and the muscles related to swallowing) [13], which is occasionally detected in older adults and is related to physical deterioration, inadequate nutritional management, and cognitive decline [14]. Perhaps multicomponent assessments, including assessments of physical, social, and psychological problems, appropriate rehabilitation, and nutrition management may be necessary for treatment [14]; however, this requires an early diagnosis of sarcopenia. In Nakahara et al. [15], we clarified the assessment of sarcopenia and cachexia between different occupations but when these items were assessed remains unclear. In Japan’s rapidly aging society, the number of older adults with sarcopenia, nutritional deficiency, frailty, and disability is increasing at an alarming rate [16]; therefore, it is important to assess physical functions and nutritional status and to screen patients at risk early. Aging is a normal physiological process, though the appearance of aging can be delayed by measurable physical activity that is carried out early and continued regularly, and also by healthy nutrition [17,18,19]. The reduction in physiological aging processes will have an impact on the physical fitness of the elderly [19,20]. Seniors aged >60 who can maintain a good physical condition will inhibit the degenerative process but, at the same time, there is evidence that there are many seniors in the community who are experiencing physical decline due to not maintaining a fit physique [21].

The process of nutrition care [22,23] and nutrition management requires a shared understanding among health care workers from different occupations. Based on these, Wakabayashi [16] suggested the provision of high-quality nutritional care using the process of nutritional rehabilitation for people with disabilities and frailty, through clinical hypercaloric nutrition. As a first step in the prevention and management of pancreatic cancer cachexia, nutritional support should be provided through counseling and oral nutritional supplements to prevent and minimize the loss of lean body mass and systemic inflammation, determined by a combination of host cytokines [9].

While sarcopenia is a nutritionally related disease, malnutrition and cachexia are nutritional disorders that share the common feature of low fat-free mass [3,24]. Applying current definitions in clinical practice is still a challenge for healthcare professionals and the potential for misdiagnosis is high. This is of particular concern in the subgroup of older people with cancer, where sarcopenia, malnutrition, and cancer cachexia are highly prevalent and may overlap or occur separately [24]. Knowledge of the epidemiology of malnutrition/cachexia/sarcopenia can help manage these complications early in treatment, potentially impacting the patient’s quality of life, treatment intensity, and disease outcome [25,26].

Osteoporosis, sarcopenia, and cachexia impose a huge cost burden on health and social services in countries [27]. The identification of the first signs of cachexia through non-invasive, simple methods can bring advantages in the management of this disease, with the possibility of intervening in the underlying disease. If cachexia is in the advanced phase, none of the diet-therapeutic interventions seem to reach the desired goal [28,29,30]. This study aims to clarify the correlation between BIA-measured parameters, clinical parameters, and other nutrition-related assessments. They also look for explanations for the massive loss of muscle mass and lean mass despite the hypercaloric diet that was applied. In addition, this study intends to evaluate the correlation of the sarcopenic paraclinical and nutritional parameters with cachexia.

## 2. Materials and Methods

### 2.1. Study Design and Setup

Osteoporosis, being the underlying condition of the research study, was checked by DEXA osteodensitometry. The number of subjects was 200, of which 15 (7.5%) were diagnosed with sarcopenia/cachexia. Based on this, the cohort was divided into 2 groups: group 1 (those with sarcopenia/cachexia made up of 15 people) and group 2 (without sarcopenia/cachexia made up of 185 patients). The mean T-score value at the beginning of the cohort research period was −2.84 ± 0.29, with insignificant differences between research groups (χ^2^ = 0.089). At the end of the research, the average T-score value was −2.86 +0.29 and *p* = 0.017 (χ^2^ = 0.017). The paraclinical analyses for the assessment of malnutrition, hemoglobin, and albumin were performed in the analysis laboratory by enzymatic, colorimetric, and spectrophotometric methods and an enzyme immunoassay (ELISA). Inclusion criteria: patients with osteoporosis (T-score < −2.5) and those with low physical performance, SPPB score < 9. Exclusion criteria considered were as follows: T-score > −2.5, history of malignancy, articular ankylosis, organ failure, and refusal of patients to participate in the study.

### 2.2. Ethical Considerations

The study was carried out during 2020–2021, at the Center for Medical Treatment and Rehabilitation, Băile 1 Mai, Hotel Ceres, Bihor, Romania, with the agreement of the institution’s management in accordance with the Declaration of the World Medical Association of Helsinki (3918 of 16.11.2020). Participation in the study was voluntary.

### 2.3. The Method of Determining the Bone Density (DEXA)

All patients evaluated had their bone density determined with an osteodensitometer (MEDIX 90, Gallarques de Montueux, France). Bone mineral density (BMD) was measured at the femoral neck and L2-L4 lumbar spine. Based on osteodensitometry measurements, osteoporosis is defined by T-score values lower than −2.5; osteopenia is characterized by T-score values between −1.1 and −2.5.

To follow the evolution of bone-mass loss, we classified the T-score values as follows: OP1 with T-score values between −2.5 and −2.8, and OP2 with values lower than −2.8.

### 2.4. Method for Determining Maximum Voluntary Strength

Physical performance was determined using the SARC-F and SPPB tests. Dynamometric measurements were performed to determine the maximum voluntary muscle strength in the right upper limb, as well as the left one.

### 2.5. Method for Determining Body Analysis with BIA

The clinical evaluation was performed with a body bio-electrical impedance analyzer (Tanita MC780MA, Tokyo, Japan), and the evaluation of results using medical software (GMON medical software, Chemnitz, Germany). BIA body analyzers are WPHNA (World Public Health Nutrition Association) approved devices that determine body composition with high accuracy. The margin of error is 0.1 kg. The analysis is done in the morning on an empty stomach, the patient climbs on the scale with bare feet, and grabs 2 devices. A low-frequency electric current is introduced into the feet and it is picked up from the hands. The sequential analysis looks not only at body composition but also at the distribution of fat and muscle mass in each body compartment.

### 2.6. Sarcopenia Diagnosis Algorithm and Determination of Sarcopenic Index

In this study, we applied the diagnostic algorithm for sarcopenia suggested by the European Working Group on Sarcopenia in Older People (EWGSOP) in 2019. (Figure 1). It is a simple and quick diagnostic algorithm that can be used in clinical practice.

### 2.7. Parameters

Paraclinical assessments were performed to support the diagnosis. Establishing the diagnosis of osteoporosis was performed by determining osteodensitometry. To confirm the static vertebral disorders, radiographs of the spine (front and profile) were performed. Paraclinical analyses were performed in the analysis laboratory by enzymatic, colorimetric, and spectrophotometric methods and an immuno-enzymatic test. We monitored 2 metabolic parameters, namely, albumin and hemoglobin, at the beginning and end of the research period.

### 2.8. Diet Therapeutic Method

All patients underwent a clinical osteoporosis-specific diet therapy to treat or prevent sarcopenia. The diet therapy applied was a hypercaloric one (hyperprotein, hyperglycemic, normolipidic) with an excess between 10–25% of the macronutrient intake.

### 2.9. Statistical Analyses

The data obtained were analyzed in the statistical program SPSS 20 (New York, NY, USA) by statistics (ANOVA), post-hoc analysis, the Chi-square test, and inferential statistics (Student t-test), and the 3 research groups were compared with the Bonferroni test. Correlations between parameters were checked using Bravais-Pearson tests and pairwise correlations of the samples.

## 3. Results

### 3.1. Demographic Description

The cohort of 200 patients, most of whom came from an urban environment (52%), is described in Table 1, with non-significant differences (*p* = 0.572). The distribution of the study cohort in terms of gender was as follows, women from the urban environment represented 40%, which is higher than those from the rural environment, representing only 33%. Men from urban areas represented 12%, and from rural areas, 15%.

The average age of the cohort was 67.43 ± 7.94 years, demonstrating a specific age of the study, as this age is more prone to the development of osteoporosis and its adjacent diseases. Both at the level of the cohort and in each group, the age of the patients ranged between 46 and 82 years with significant age differences (*p* = 0.001), the highest average age being in group 1 at 68.03 ± 8.09, presented in Table 1.

More than 50% of patients were between 61–70 years old (52.20%), the average age being 66.94 years.

### 3.2. Statistical Analysis of Research Parameters

Patients were divided into two groups; the sarcopenic index group <6 in women and <7 in men. Among the monitored clinical and paraclinical parameters were BMI, fat mass, osteoporosis, sarcopenic index, albumin, and hemoglobin, which are shown in Table 2 initial values, and Table 2 final values. Both groups were subjected to a hypercaloric (hyperprotein, hyperglycemic, normolipidic) diet therapy with an excess of between 10–25% of the macronutrient intake.

After one year of research, a significant decrease in BMI (*p* = 0.020) and fat mass (*p* = 0.001) was observed at the cohort level, with significant differences between the research groups (*p* = 0.001) in both parameters. Osteoporosis (0.298) and the sarcopenic index (0.901) did not change with statistical significance at the cohort level but did change significantly at the research group level (osteoporosis *p* = 0.001, sarcopenic index *p* = 0.034). The albumin change was statistically significant (*p* = 0.017) and for hemoglobin (*p* = 0.008) at the cohort level and following each research group, a significant decrease was observed.

At the same time, a more pronounced decrease in fat mass was observed in men, correlated with a higher value of the sarcopenic index. These results lead to higher BMI in men, so women are more affected by the loss of muscle mass and fat mass, with the risk of sarcopenia progressing towards cachexia.

### 3.3. Correlations

The Pearson correlations of the research parameters in group 1 are presented in Table 3. A strongly positive, statistically significant relationship (*p* < 0.05) between BMI and age was recorded. The younger the patient was, the more the BMI decreased. Malnutrition was underlined by the strong negative correlation of BMI with albumin (*p* = 0.001) and hemoglobin (*p* = 0.001). Osteoporosis also increased with age (*p* = 0.021). As the fat mass difference increased, the albumin difference also increased; therefore, the more fat mass that was lost, the more the albumin value decreased. Albumin is significantly lower in younger people (*p* = 0.001), and the lower the albumin, the lower the hemoglobin (*p* = 0.006).

Pearson’s correlation shows there is a strong relationship between the two parameters, i.e., the study groups and fat mass and hemoglobin. In the group with sarcopenia/cachexia, a reduction in fat mass and hemoglobin was observed. The estimated increase in fat mass in group 2 is 59.3%, represented by R^2^, and a 25.3% increase in hemoglobin was also present in group 2, shown in Table 4 and Figure 2. The ANOVA test shows whether the model is statistically significant.

## 4. Discussion

Along with the development of body analyzers based on bioelectrical impedance (BIA), the possibility to measure the amount of muscle mass noninvasively and with minimal errors has also been developed. Some innovations from the year 2015 gave the possibility to determine the quality of the muscles, following the sarcopenic index [3]. It has proven very useful in the follow-up of the frailty syndrome, being able to correlate with malnutrition [31]. In the present study, the sarcopenic index was followed, both as a diagnostic component and as a parameter, correlated with the regression study also being followed in the estimation of risk. A meta-analysis conducted on 68 studies (2021), which included a number of 98,502 patients, looked at the association of socio-demographics, behavioral factors (nutrition, smoking, physical activity), and associated diseases (including osteoporosis) with sarcopenia in the general population. Only six studies referred to the decrease in bone density and support the fact that OP is a risk factor associated with sarcopenia in the elderly population [32]. Sarcopenia is significantly related to osteopenia and osteoporosis, as demonstrated in a study of 3077 volunteers over the age of 65, regardless of the associated pathology (Lee et al., 2021). Sarcopenia was determined in 1230 patients (39.9%), of which 41.8% were men (*p* = 0.133). Osteopenia was present in 1402 subjects evaluated (44.0%), of which 53.1% (*n* = 750) were men (*p* < 0.001); osteoporosis was diagnosed in 1156 patients (39.9%), of which 990 were women (59.9%, *p* < 0.001), with a higher prevalence in men (54.9% vs. 67.9% in women) [33]. None of the studies showed an association between smoking and sarcopenia [32].

The results of the present study show a statistically significant difference in the sarcopenic index, which evaluates muscle strength, assistance in walking, rising from a chair, climbing stairs, and falls between the two studied groups. A 2020 meta-analysis found that markers of malnutrition (15 biological markers) had poorer clinical outcomes in those with cachexia cancer. In our study, we tracked six markers that showed poorer final results in group 1 for five of the monitored parameters, and the 6th, BMI, did not worsen but did not show significant improvement either [34]. Additionally, in this study food was followed, and it was found that the decrease in nutritional intake was correlated with a higher mortality rate [34]. We increased the nutritional intake but did not observe significant improvements in the research parameters. Obesity is correlated with a high level of inflammation and is considered a systemic inflammatory disease [35]. Overweight and obese children and adults have elevated serum levels of C-reactive protein, interleukin-6, tumor necrosis factor-alpha, and leptin, which are known markers of inflammation and strongly associated with cardiovascular risk factors, including cardiovascular and non-cardiovascular causes. These complications of obesity are accentuated with increased inflammation [36]. Precision nutrition, i.e., personalized down to the molecular level including genetic aspects, has a significantly positive influence on health, states a study from 2017 [37]. Another study conducted over 18 months on 680 patients over 70 years of age, regarding the association between bed rest and functional decline, demonstrated a relationship between time spent on bed rest and the size of functional decline with decreased mobility, physical activity, and social, in the performance of ADLs [38]. The incidence of moderate sarcopenia among bedridden US men is 71% and 42% among women (≥65 years) [39]. In our study, the overall incidence of sarcopenia was 20% in adults aged over 70 and over 50% in institutionalized personnel over 80. The severe form of sarcopenia, which is responsible for severe disability, is more frequently found among men (17% vs. 11% in women). The identification of sarcopenia through dynamometric measurement is an insufficient method of diagnosis, this has been proven since 2012 [40], but is indispensable to determine the decrease in muscle strength during that period. The dynamometric analysis concluded that 58% of the patients showed a decrease in muscle strength.

Blood analyses specific to severe malnutrition include several parameters besides hemoglobin, albumin, and osteoporosis, according to a study done in the Clinical Unit of Nutrition-Eating Disorders in the Raymond Poincaré Hospital, France [41]. They looked at anorexia nervosa related to reduced food intake. In our study, the food intake is hypercaloric but signs of malnutrition reflected in paraclinical parameters are also present.

Cachexia is a wasting disorder that accompanies many chronic diseases, including cancer, and results from an imbalance of energy requirements and energy expenditure [42]. In cancer cachexia, tumor-secreted factors and/or tumor-host interactions cause this imbalance, leading to the loss of adipose tissue and skeletal and cardiac muscle, which weakens the body [42]. The present study has several limitations. First, our findings were obtained from a cross-sectional study; there were no assessments of more detailed physical performance and the presence of sarcopenia by MRI, hormonal determinations (testosterone, estrogen), or osteodensitometry determinations at the end of the study.

One of the study’s limitations is also its strength, that is, the use of the BIA body analyzer. This is a limitation, being non-invasive with an accuracy of ±0.1 kg only in the above-mentioned conditions. If the measurements are made during the day, the values can be modified according to water consumption [43] or the amount of food consumed. The strength of the body analyzer is the accessibility and stability of the analysis under the conditions recommended by the manufacturer. Another limitation is the tracking of the food intake, particularly the protein intake. It can’t be tracked with incredible accuracy because the protein supplementation recommended isn’t strictly monitored. The incidence of diabetes was also not evaluated.

## 5. Conclusions

In conclusion, women presented poorer results and a higher incidence of cachexia than men. Nutritional interventions were better utilized in men who had a slower progression of the disease, in contrast to women who did not assimilate nutrients as well; this was reflected in the paraclinical parameters at the end of the research period (albumin 3.91 ± 1.37 in women and 4.02 ± 0.98 in men; hemoglobin 11.47 ± 1.22 in women and 13.79 ± 1.47 in men). The estimated risk of fat-mass loss and decreased hemoglobin is higher in the sarcopenia/cachexia group.

The hypercaloric diet did not positively influence the evolution of cachexia, so the approach based only on diet therapy in the case of cachexia was insufficient.

## Figures and Tables

**Figure 1 geriatrics-07-00102-f001:**
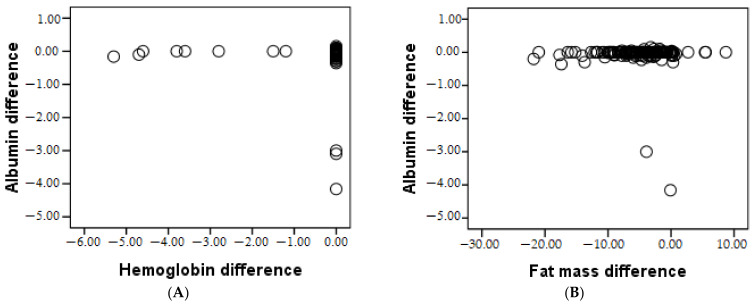
The correlation between albumin and hemoglobin (**A**), and fat mass (**B**), respectively.

**Figure 2 geriatrics-07-00102-f002:**
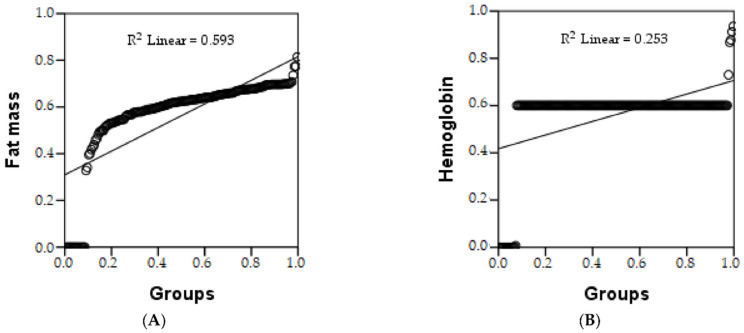
Linear regression of: fat mass (**A**), and hemoglobin (**B**), depending on groups.

**Table 1 geriatrics-07-00102-t001:** Distribution of cases according to the environment of origin, and cases according to age.

Environment of Origin		Total		Group 1		Group 2	
		N	%	N	%	N	%
Rural		96	48.0	8	53.3	96	51.9
Urban		104	52.0	7	46.6	89	48.1
Total		200	100.0	15	7.5	185	92.5
Age	<60 years	36	18.0	5	33.3	31	16.7
	61–70 years	104	52.0	7	46.6	97	52.4
	>70 years	60	30.0	3	20	57	30.8
Min-max		46–82					
Mean age		67.43 ± 7.94		68.03 ± 8.09		66.91 ± 7.81	

N—total number of patients; Min-max—minimum and maximum value.

**Table 2 geriatrics-07-00102-t002:** The initial values of the research parameters at the level of the cohort and in both groups.

Parameters	Total				Groups							
					Group 1				Group 2			
	Initial		Final		Initial		Final		Initial		Final	
	Mean	SD	Mean	SD	Mean	SD	Mean	SD	Mean	SD	Mean	SD
BMI	28.44	5.10	28.48	5.09	24. 02	2.76	24.12	2.80	27.68	4.37	28.83	5.08
Fat mass	30.64	7.41	27.04	6.61	16.55	2.66	15.05	3.40	31.99	6.22	28.01	5.80
Osteoporosis	−2.83	0.29	−2.85	0.29	−2.79	0.31	−2.86	0.29	−2.79	0.26	−2.84	0.30
Sarcopenic index	8.12	1.42	8.11	1.44	6.14	0.62	6.11	0.60	8.28	1.34	8.27	1.37
Albumin	4.06	1.02	3.99	1.03	3.69	1.29	3.68	1.28	4.11	0.99	4.01	1.01
Hemoglobin	12.24	1.77	12.10	1.65	13.43	2.02	12.85	1.82	12.15	1.82	12.04	1.62

SD = Standard Deviation.

**Table 3 geriatrics-07-00102-t003:** Pearson correlation of the research parameters in group 1.

Pearson Correlation		Age	Gender	Albumin	Hemoglobin
BMI difference	r	**0.558 ***	−0.161	−0.848 **	−0.831 **
	Sig.	0.031	0.566	0.000	0.000
Osteoporosis difference	r	0.259	**−0.590 ***	−0.156	−0.342
	Sig.	0.351	0.021	0.579	0.212
Sarcopenic index difference	r	−0.194	0.080	−0.016	−0.331
	Sig.	0.488	0.776	0.954	0.229
Albumin difference	r	**−0.790 ****	0.228	1	0.674 **
	Sig.	0.000	0.413	1	0.006
	N	15			

r = Pearson coefficient; Sig. = statistical signification; N = number of patients; ** Correlation is significant at the 0.01 level (two-tailed); * Correlation is significant at the 0.05 level (two-tailed).

**Table 4 geriatrics-07-00102-t004:** Regression table of quality-of-life parameters according to research groups.

Parameters	Correlate	R^2^ (%)	r	β1	β2	t	F
Fat mass	Groups	59.30	0.201 **	−0.012 *	−0.201 *	62.822 **	7.140 **
Hemoglobin		25.30	0.173 **	0.062 *	0.173 *	103.059 **	6.085 **

R^2^ = R square; β1 = unstandardized coefficient; β2 = standardized coefficient; t = coefficient of correlation; F = ANOVA coefficient; ** Correlation is significant at the 0.01 level (two-tailed); * Correlation is significant at the 0.05 level (two-tailed).

## Data Availability

Not applicable.

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
