# Peer review of "Evaluation of Geriatric Sarcopenia and Nutrition in the Case of Cachexia before Exitus: An Observational Study for Health Professionals"

_geriatrics, 2022, doi:10.3390/geriatrics7050102_

Round 1

Reviewer 1 Report

The manuscript with the title "Evaluation of geriatric sarcopenia and nutrition in case of cachexia before exitus, an observational study for health professionals" aims to clarify the correlation between BIA-measured parameters, clinical parameters and other nutrition-related assessments. However, there are many major drawbacks in this study the affect the scientific soundness and validity of this study. 

1. The introduction need to be improved to better described the knowledge gap and justified the need of study

2. The objective of the study is not clear and the methods and findings of the study did not really answer the study objective

3. Methodology is poorly described. No inclusion and exclusion criteria is stated. The authors aims to study the geriatric sarcopenia, but include osteoporosis as one of the main measurements of the study. The methods on how to perform the BIA analysis is not described. 

4. The authors mentioned "The paraclinical analysis were carried out in the analysis laboratory using enzymatic, colorimetric and spectrophotometric methods and enzyme immunoassay (ELISA)." what are the parameters that being measured using this method? The authors did not state it clearly in the manuscript.

5. The data presentation is poor. Table 1 and 2 should combine together, together with other sociodemography data to better describe the characteristics of the participants.

6. "Both groups were subjected to a hypercaloric (hyperprotein, hyperglycemic, normolipidic) diet therapy with an excess of between 10-25% of the macronutrient intake" So there is an nutrition intervention for both group? I though this is an observational study, as stated in the title of the manuscript. What is the purpose of giving nutritional intervention? It is not clearly stated in the objective and methodology.

7. Is there any approval from human ethics committee for conducting this research? please provide the ethics approval code/prove.

8. Table 3 and 4 should combined as well, to make better comparison before and after 1 years of experiments.

9. the findings of the study did not answer the objective of the study. The authors also failed to discuss and correlate the findings to the main aims of the study

Author Response

Firstly, we, the authors of the present manuscript wish to thank you for thoughtful commentary you have provided to improve the quality of the paper. I am very grateful for the time and effort you have devoted to this task. We have extensively revised my manuscript according to the recommendations. All changes in the text and the new figures that we have redesigned are highlighted. Please, see the point-by-point answers to your comments below. All correction was highlighted in the manuscript.

Reviewer 2 Report

Remarks:

1. The summary should be corrected. The summary should include - research problem, research goal, method, sample selection.

2. The article should precisely describe the research methods and the selection of the test sample. In detail.

3. In the discussion, comparisons with the research of other authors should be broadened.

4. In the discussion of the results, one should refer to practice more broadly.

Author Response

(The authors gave the same response as above.)

Round 2

Reviewer 1 Report

The authors answered most of the comments. There are significant improvement after the revision. 

Minor comment:

Table 2 is redundance since the authors have combined the data in table 1. Please remove it.